# Research

behaviour, cognition, neuroscience

individual difference, spatial biases, spatial localization, visual acuity, size perception

**Author for correspondence:**
Zixuan Wang
e-mail: zixuan@berkeley.edu

# Idiosyncratic perception: a link between acuity, perceived position and apparent size

Zixuan Wang[1], Yuki Murai[1,4,5] and David Whitney[1,2,3]

[1]Department of Psychology, [2]Vision Science Program, and [3]Helen Wills Neuroscience Institute, University of California, Berkeley, CA, USA
[4]Japan Society for the Promotion of Science, Chiyoda-ku, Tokyo, Japan
[5]Graduate School of Frontier Biosciences, Osaka University, Osaka, Japan

ZW, 0000-0002-8001-2233

Perceiving the positions of objects is a prerequisite for most other visual and visuomotor functions, but human perception of object position varies from one individual to the next. The source of these individual differences in perceived position and their perceptual consequences are unknown. Here, we tested whether idiosyncratic biases in the underlying representation of visual space propagate across different levels of visual processing. In Experiment 1, using a position matching task, we found stable, observer-specific compressions and expansions within local regions throughout the visual field. We then measured Vernier acuity (Experiment 2) and perceived size of objects (Experiment 3) across the visual field and found that individualized spatial distortions were closely associated with variations in both visual acuity and apparent object size. Our results reveal idiosyncratic biases in perceived position and size, originating from a heterogeneous spatial resolution that carries across the visual hierarchy.

## 1. Introduction

Accurately registering the locations of objects is a critical visual function. Most other perceptual functions, including pattern and object recognition, as well as visually guided behaviour, hinge on first localizing object positions. Position perception is generally assumed to be dictated by retinotopic location, and that may explain a lot of the variance in perceived position. However, perceived position can be biased due to various external factors, such as overt attention [1], motion [2] and saccadic eye movements [3]. The impact of these factors can be significant, especially considering the spatial scale at which object recognition and visually guided action happen. A 0.5-degree shift in the location of a pedestrian or car crossing a freeway could result in a catastrophic collision. The scale at which perception and action needs to operate is often very fine, and many factors bias perceived position at a scale that is behaviourally relevant.

In the absence of these external factors, perceived position is often assumed to be uniformly dictated by retinotopic position. However, a recent study challenges this belief and demonstrates that people mislocalize objects idiosyncratically and consistently even without apparent change in the environment [4]. The unique biases in object locations were shown to be stable across time when tested after weeks or months, indicating a stable perceptual fingerprint of object location.

Why do people perceive idiosyncratically biased object locations in different parts of the visual field and what are the perceptual consequences of it? Here, we test the possibility that variations in spatial resolution across the visual field might cause the spatial distortions in perceived position. Many researchers have shown that visual acuity varies across the visual field [5–7]. Because many models of localization depend implicitly or explicitly on the underlying

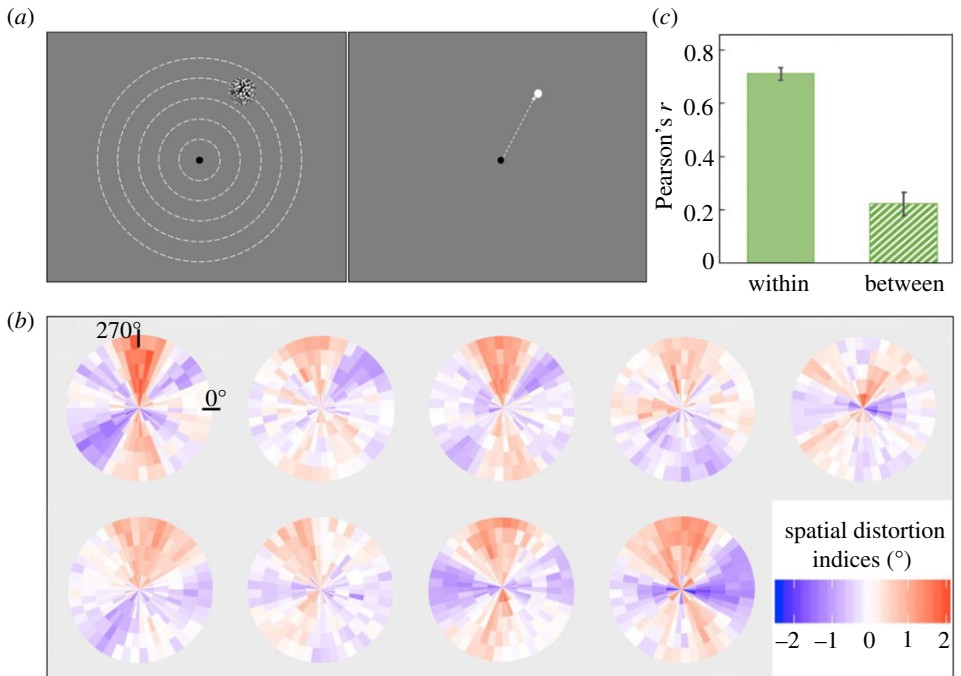

**Figure 1.** Experimental paradigm and results for Experiment 1. (*a*) Left: observers fixated at the centre and a target was displayed briefly at one of five possible eccentricities (depicted by dashed lines, which were not visible in the experiment). Right: after the target disappeared, observers moved the cursor to match the target's location. (*b*) All nine individual observers' spatial distortion indices plotted as distortion maps. The colour gradient represents the degree of distortion, with blue indicating contracted visual space and red for expanded. See electronic supplementary material, figure S1 for grey-scale luminance-defined distortion maps. (*c*) Averaged within versus between-subject correlation calculated by bootstrap procedures (see Experiment 1, Method). There was significantly higher within-observer agreement than between-observer agreement, indicating that each observer had a unique pattern of spatial distortions. The error bars represent the bootstrapped 95% CI. (Online version in colour.)

resolution and homogeneity of spatial coding [1,8–10], it is conceivable that the inhomogeneity in visual acuity could result in an inhomogeneous visual space representation, consisting of areas of contraction (sinks) and expansion (sources). Mislocalization would be one of the natural perceptual consequences of these inhomogeneities.

A further prediction is that if individual observers have inhomogeneous visual acuity and consequential distorted representations of visual space, the biases might be carried along with the visual system so that object representations and appearance may also vary in a predictable and related way. To test this, we also measured whether the perceived size of objects varies at individualized perceptually contracted or expanded regions of visual space.

## 2. Experiment 1: idiosyncratic visual space distortion

Kosovicheva & Whitney [4] demonstrated that observers have stable and idiosyncratic patterns of mislocalization at different polar angles in the visual field. We hypothesized that this mislocalization pattern reflects distinct distortions of visual space and that it should be observed from the fovea to the periphery (not just at one eccentricity). The purpose of Experiment 1 was to identify whether there are idiosyncratic spatial distortions across the visual field.

### (a) Method
#### (i) Participants
Nine observers (3 females, 2 authors, age range: 19–33) participated in this experiment. All subjects were experienced

psychophysical participants, and all but the two authors were naïve to the purpose of the study. All subjects reported to have normal or corrected-to-normal vision. Procedures were approved by the Institutional Review Board at the University of California, Berkeley.

#### (ii) Stimuli
Stimuli were presented on a 19-inch gamma-corrected Dell P991 CRT monitor (Dell, Round Rock, TX; $1024 \times 768$ pixels resolution, 100 Hz refresh rate). To minimize any off-screen reference (i.e. any visible references outside of the computer monitor including the difference between the monitor frame and the experiment room), the monitor frame was covered by black tape. Visual stimuli were generated using MATLAB (The MathWorks, Natick, MA) and Psychophysics Toolbox (Version 3) [11] and the experiment program was run on an Apple Macintosh computer (Apple Inc., Cupertino, CA). Observers viewed the stimuli binocularly at a distance of 40 cm using a chin rest.

We used noise patches as targets for the localization task (figure 1*a*). Each noise patch contained random black ($<0.01\ \mathrm{cd\ m^{-2}}$, measured by Minolta LS110 Luminance Meter) and white ($92.6\ \mathrm{cd\ m^{-2}}$) squares (each square was $0.1 \times 0.1$ degrees of visual angle [d.v.a.]). Noise patches were enveloped with a two-dimensional Gaussian contrast aperture (standard deviation: 0.75) and only visible within a circular aperture with a radius of 1.22 d.v.a. Noise patches were shown on one of 5 invisible isoeccentric rings (eccentricity: 2, 4, 6, 8, 10 d.v.a.) and one of 48 angular positions equally distributed with a separation of 7.5° on each ring, which resulted in a total number of 240 possible locations. Angular locations from 0° to 360° corresponded to positions

starting from the right of the fixation, moving clockwise. The four exactly vertical and horizontal positions at each eccentricity were included.

### (iii) Procedure

Observers were instructed to maintain fixation throughout the experiment. On each trial, a black ($<0.01$ cd m$^{-2}$) 0.3-d.v.a. diameter fixation dot was presented at the centre of a grey (48.03 cd m$^{-2}$) background on the screen. After 1000 ms, a noise patch appeared at a pseudo-randomly chosen location among all 240 locations for 50 ms. Upon the offset of the noise patch, the fixation dot changed to dark grey (30.4 cd m$^{-2}$) and 500 ms later, a white (92.6 cd m$^{-2}$) 0.45 d.v.a. diameter response dot representing the location of the cursor was superimposed on the fixation and participants freely moved the dot using the mouse to match the location of the noise patch centre. The position of the cursor when participants clicked the mouse was recorded as the reported location.

In the experiment, each target location was tested 12 times, so there were 2880 trials in total (12 repetitions × 48 angular locations × 5 eccentricities). The whole experiment was separated into six sessions (2 repetitions for every location per session, random sequence within session). The time interval between every six sessions was on average 1.3 days (standard deviation: 1.6 days).

### (iv) Data analysis

For each observer and each session, we first calculated the polar angles of the cursor locations reported by the observer. Then for each target location, the two reported locations from the two trials were averaged within each session. Thus, there were six sessions of averaged reported locations. Within each session, the average locations were grouped by five eccentricities with each consisting of 48 isoeccentric reported angular locations.

For each session and at each eccentricity, the 48 reported locations were transformed into 48 visual space distortion indices. Since any two physically adjacent target locations at the same eccentricity were separated by 7.5° polar angle, if the angular distance between the two adjacent reported locations in the same session was larger than 7.5°, then the area between them was effectively a region of expanded visual space. On the other hand, if the distance was smaller than 7.5°, the visual space between them was effectively compressed. Thus, when the physical target distance (i.e. 7.5° polar angle) was subtracted from the reported distance (i.e. difference in perceived locations), it yielded a visual space distortion index in degrees of polar angle for each location. A more positive index refers to increasing visual space expansion and a more negative index refers to larger visual space compression. Zero means no distortion. These resulting 48 distortion indices at each eccentricity within each session were smoothed using a simple moving average method with a window of 45° polar angle. The smoothing was to better characterize a continuous change of distortion across space and to compensate for the discrete spatial sampling given that only 48 locations were tested at each eccentricity. The smoothed distortion indices were averaged across the six sessions at each location (shown as individual distortion map in figure 1b).

To quantify the idiosyncrasies of the distortion indices, we compare the within-observer to between-observer consistency using a nonparametric bootstrap method [12]. The within-observer consistency is defined as the similarity of the distortion indices across sessions. On each iteration, for every observer, three random sessions were sampled without replacement from all of the six sessions and the spatial distortion indices for each location were averaged across the three sessions as one bootstrapped half. The three remaining unsampled sessions were averaged and formed the second half. We z-scored the distortion indices at each eccentricity within each half in order to remove the effect of eccentricity and then correlated the two halves. The Pearson's r value for each observer was transformed into Fisher z value before averaging across observers and the averaged Fisher z was transformed back into Pearson's r value (Fisher transformation, used for all procedures that required averaging correlation values). This procedure was repeated 1000 times to estimate a 95% bootstrapped confidence interval (CI) for within-observer consistency. Between-observer consistency was estimated similarly. On each iteration, one of the two halves from one observer was correlated with one half from another observer. All possible pairwise between-observer correlations were averaged together. This procedure was also repeated 1000 times to estimate a 95% bootstrapped CI for between-observer consistency.

To evaluate the significance of the within-observer and between-observer correlations, we also generated permuted null distributions that showed the expected chance correlations from two uncorrelated and permutated halves of the distortion indices. On each iteration, for every observer, all distortion indices were randomly split into two halves (as described previously). One of the two halves was rotated by a random number of positions at each eccentricity to shift the distortion indices away from its original physical positions while simultaneously preserving the spatial relationships between adjacent isoeccentric target locations. The rotated half was then correlated with the other unchanged half from the same observer to estimate a within-observer null correlation. Then, the null correlations were averaged together. This procedure was repeated 10 000 times to estimate a within-observer permuted null distribution. For the between-observer null distribution, on each iteration, the rotated half was correlated with an unchanged half from another observer and all pairwise null correlations between observers were averaged together. This procedure was also repeated 10 000 times. The mean within-observer and between-observer empirical correlations obtained in the bootstrap analyses described above were compared to these null distributions.

To quantify the unique contributions of the distortion indices within each individual rather than a common distortion pattern between observers, we fitted linear regression models and compared the variance explained by different models using a bootstrap test in R [13]. On each iteration, we first fitted a full model which was formalized as

$$\mathrm{DI} \sim \beta_0 + \beta_1 * \mathrm{self} + \beta_2 * \mathrm{others}.$$

The dependent variable DI (*distortion indices*) for every target location was calculated by randomly sampling half of the six sessions without replacement and averaging the distortion indices across the three sampled sessions. Then the three remaining unsampled sessions were also averaged to

represent the observer-specific spatial distortion pattern (*self*). The other predictor (*others*) was the distortion indices averaged across the remaining observers, using only half of the sessions from every observer to minimize the signal-to-noise differences between the two predictors. To estimate how much variance of each observer's distortion indices can be explained by their own distortion pattern (*self*) versus the other observers' averaged distortion pattern (*others*), we also fitted two other models, the within-observer and the other-observer models. The within-observer model was formalized as

$$DI \sim \beta_0 + \beta_1 \times \text{self},$$

and the other-observer model was expressed as

$$DI \sim \beta_0 + \beta_1 \times \text{others}.$$

Unique variance explained by *self* (i.e. the distortion indices within each observer) was estimated by subtracting variance explained by the other-observer model from that explained by the full model. Variance explained by *others* (i.e. the averaged distortion indices across other observers) was estimated by subtracting variance explained by the within-observer model from the full model. We also estimated shared variance between *self* and *others* by subtracting the unique variances of each of them from the explained variance of the full model. Note that since the shared variance is the common contribution between observers themselves and others, it essentially represents a between-observer similarity in the spatial distortion indices. We repeated this procedure 1000 times to compare the unique variance explained by within-observer or other-observer distortion indices, or the shared variance between observers.

## (b) Results

The bootstrapped within and between-observer similarity is shown in figure 1c. The average bootstrapped within-observer correlation ($r = 0.71$) was significantly higher than the null correlation expected by chance ($p < 0.001$, permutation test). This reveals consistent spatial distortion patterns within individual observers. The mean bootstrapped between-observer correlation ($r = 0.22$) was significantly higher than chance ($p < 0.001$, permutation test). We further found that within-observer similarity is significantly higher than between-observer similarity ($p < 0.001$, bootstrap test), suggesting that each individual observer has their own unique spatial distortions that are consistent within themselves and distinguished from other observers (figure 1c), consistent with a previous study [4]. To quantify the unique contributions of within-observer versus between-observer effects, we also fitted linear regression models and compared the unique variance explained by distortion indices within each observer versus averaged distortion indices across other observers, as well as the shared variance, using a bootstrap procedure (see Data analysis). Results showed that distortion indices within observers (*self*) on average uniquely explained 76.76% of the variance in the full model. Put simply, this means that a particular observer predicts their own pattern of distortions very well. The unique variance that cannot be explained by the observer themselves and can only be explained by the distortion indices from other observers (*others*) was less than 0.1%. This is not surprising:

it simply means that other observers' judgements do not have any explanatory power beyond what is already explained by one's own pattern of distortion (i.e. it should be zero). The shared variance between *self* and *others* on average explained 23.16% of the variance in the full model. This shared variance is akin to the between-subject similarity in figure 1c. The unique variance explained by distortion indices within each observer was significantly larger than both the averaged distortion indices across other observers ($p < 0.001$, bootstrap test, Bonferroni corrected $\alpha_B = 0.025$) and the shared variance between these two predictors ($p < 0.001$, bootstrap test, $\alpha_B = 0.025$). Since the shared variance between *self* and *others* captures the shared contributions from every observer versus all other observers, this is essentially a between-observer effect. The regression models show that idiosyncratic observer-specific biases are the major contributor to the spatial distortions, rather than a common spatial bias among observers.

## 3. Experiment 2: associate individual distortion fingerprints with visual acuity

In Experiment 1, we demonstrated that individual observers have unique visual space distortions. Where do these idiosyncratic spatial biases emerge? Given that previous studies have revealed substantial heterogeneity in visual acuity across the visual field [5–7], it is plausible that the spatial distortions emerge as a consequence of this. Therefore, in Experiment 2, we measured Vernier acuity [14] at different spatial locations to assess the potential association between spatial distortions and variations in visual resolution. We used Vernier acuity task because unlike other acuity measurements such as grating acuity, Vernier acuity, also called hyperacuity [15], exceeds the limits imposed by the maximal cone density on the retina, and has been shown to measure acuity at a cortical level [16].

## (a) Method
### (i) Participants
Seven observers (2 females, 2 authors, age range: 19–33) who had participated in Experiment 1 participated in the second experiment. All subjects were experienced psychophysical observers, and all but the two authors were naïve to the purpose of the study. All subjects reported to have normal or corrected-to-normal vision. Procedures were approved by the Institutional Review Board at University of California, Berkeley.

### (ii) Stimuli
Stimuli were generated by the same software and displayed on the same monitor as Experiment 1. Observers were tested binocularly with a viewing distance of 40 cm fixed by a chin rest. Two long (1.5 d.v.a.) and thin (50 arcsec) white lines (92.6 cd m$^{-2}$) oriented towards the centre of the screen were shown on a grey background (48.3 cd m$^{-2}$). On half of the trials, the outer line was positioned clockwise from the inner line and counter-clockwise for the other half of the trials (figure 2a shows the counter-clockwise condition). Five possible spatial misalignments between the two lines were deployed; the range of misalignments was customized for each observer after a practice block at the

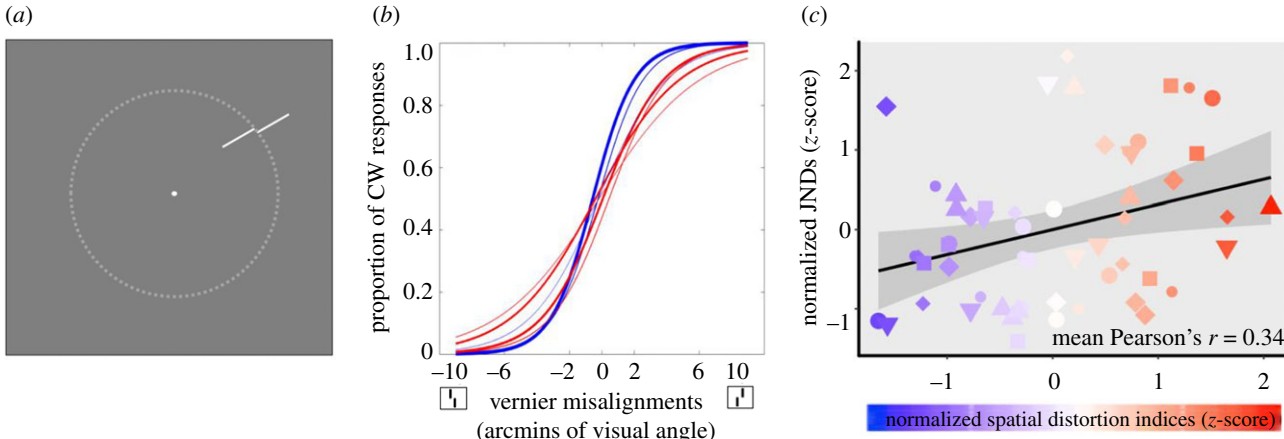

**Figure 2.** Paradigm and results for Experiment 2. (*a*) Example Vernier stimuli. Stimuli were shown on an invisible circle with a radius of 6 d.v.a. (dotted line). (*b*) Psychometric curves fitted for one representative subject. The colours of the lines correspond to locations in the visual field that had different types of distortion (blue for compression and red for expansion, as measured in Experiment 1); the weights of the lines correspond to the intensity of the distortion. (*c*) A visualization of the correlation between Vernier acuity JNDs (Experiment 2) and corresponding spatial distortion indices (Experiment 1) for all 7 observers (represented by different shapes and sizes) collapsed into a single super-subject. Blue shading indicates more contracted locations and red shading indicates more expanded locations from Experiment 1. The black line is the best-fitted linear regression line based on this super-subject data and the grey-shaded area represents the 95% CI of the linear fit. There was a significant positive correlation between the degree of visual space distortion and Vernier JNDs, indicating that better acuity was found at perceptually contracted visual space compared to expanded visual space. (Online version in colour.)

beginning of the experiment. Among all observers, the spatial misalignments were between 0.3 to 10 arcmin of visual angle. As a result, there were 10 possible Vernier stimuli (2 possible positional relationships × 5 possible spatial misalignments). The centre of the range of Vernier stimuli was located at one of 8 angular positions (45° spacing, starting with 20° polar angle), on an invisible isoeccentric ring with a radius of six d.v.a.

### (iii) Procedure

On each trial, a black (<0.001 cd m$^{-2}$) 0.3 d.v.a. diameter fixation dot was presented continuously at the centre of a grey screen to help observers maintain fixation during every trial. After 1000 ms, two Vernier lines were shown for 500 ms on a pseudo-random location selected from the eight possible positions as described above. The 10 different Vernier stimuli were presented in a pseudo-random sequence, such that every appearance was guaranteed to be displayed the same number of trials on every location. After the Vernier stimuli disappeared, observers responded by either pressing left key or right key to indicate the spatial relationship of the two lines (yes/no design). If the more eccentric line appeared to be relatively more counter-clockwise (clockwise), observers were instructed to press the left (right) arrow key. Observers were instructed to respond as accurately as possible and they had unlimited time. They were also told to fixate on the fixation dot at all times.

In the experiment, each of the 10 Vernier misalignments was repeated 20 times, resulting in 200 trials at every tested location. Since we presented stimuli at eight possible positions, there were 1600 trials in total. The whole experiment was separated into 16 blocks and each block contained 100 trials. Observers were encouraged to take a break after each block.

### (iv) Data analysis

For every Vernier misalignment at a given location, we calculated the proportions in which observers reported that the outer line was shifted more clockwise than the inner line.

Then, we fitted the proportion of clockwise responses with a logistic function using a least-squares procedure. The just-noticeable difference (JND) was estimated by taking half of the distance between the Vernier misalignments that gave 25% and 75% clockwise responses on the best-fit logistic function. This fitting procedure and JND estimation were conducted separately for each location; thus, we obtained eight JND values from each observer (see figure 2*b* for a representative subject).

To estimate the spatial distortion indices at these eight locations, for each observer, each of the eight locations tested in this experiment was rounded to the two nearest locations that had their own corresponding distortion indices in Experiment 1 at the eccentricity of 6 d.v.a. and the two distortion indices were averaged as a proxy of the spatial distortion at each of the eight locations tested in Experiment 2. We then calculated the Pearson's correlation between spatial distortions indices (Experiment 1) and Vernier acuity JNDs (Experiment 2) for every observer separately. For each of the seven observers, eight Vernier acuity JND values were correlated with the corresponding eight spatial distortion indices estimated from Experiment 1. This yielded in total 7 Pearson's *r* values, which were Fisher-transformed and averaged to estimate the mean correlation. We also performed a bootstrap procedure on these seven correlation values to test whether mean correlation value was biased by extreme observer(s). On each iteration, we randomly sampled seven correlation values with replacement from the seven empirical Pearson's *r* values and averaged the seven Fisher-transformed correlations. We repeated this procedure for 1000 times and estimated the 95% bootstrapped CI for the mean correlation among observers.

To examine whether individual differences play a role in the relationship between Vernier acuity JNDs and spatial distortions, we fitted a linear mixed effect regression on the data. The model could be expressed as

$$\text{JND}_{i,j} = \beta_0 + \beta_1 * \text{DI}_{i,j} + \gamma_i.$$

$JND_{i,j}$ was the Vernier acuity JND calculated for the $j$th location of the $i$th observer. $DI_{i,j}$ was the distortion index calculated for the $j$th location of the $i$th observer. $\gamma_i$ represented the random effect for the $i$th observer. We also constructed a simple linear model without accounting for the random effect of observers, which could be formalized as below.

$$JND_{i,j} = \beta_0 + \beta_1 * DI_{i,j}.$$

A likelihood ratio test was performed between the two models to compare the goodness-of-fit of the models with or without the random effect of observers.

## (b) Results

The change of Vernier acuity as a function of different angular locations for each individual observer is shown in electronic supplementary material, figure S2. Figure 2c visualizes the relationship between Vernier acuity and spatial distortion with individual differences removed (i.e. Vernier acuity JNDs and spatial distortion indices were z-scored within each observer and then plotted into the same figure as a 'super subject'). We are aware that any analysis based on the 'super subject' data may be subject to the problem of pseudo-replication [17], which is why we constructed a linear mixed effect regression (see Data analysis section) and calculated individual correlations for each subject (electronic supplementary material, figure S3).

Across the individual subjects, there was a significant correlation between the JNDs calculated in Experiment 2 and the corresponding spatial distortion indices at the same locations taken from Experiment 1 (mean Pearson's $r = 0.34$, 95% bootstrapped CI: [0.06, 0.56]; electronic supplementary material, figure S3). This indicates that the spatial distortion index increased with increasing JND: the locations in the visual field where acuity was better tended to be perceived as more compressed, compared to the regions of spatial expansion where acuity was coarser. Individual observer correlations are shown in the electronic supplementary material, figure S3.

The results of the linear mixed effect model also confirmed that there was a significant and positive relationship between Vernier acuity and spatial distortions, with a fixed effect coefficient of 0.63 (standard error: 0.23, $F_{1,48} = 7.59$, $p < 0.01$). This confirms the analysis above, indicating that higher acuity is associated with compression of perceived space. Importantly, compared to the model without the random effect of observers, the mixed effect model was significantly better (likelihood ratio test, $\chi^2_1 = 39.99$, $p < 0.001$). In other words, accounting for individual differences significantly increased the performance of the mixed effect model, which suggests that the association between Vernier acuity and spatial distortion is characterized by observer-specific idiosyncrasies.

# 4. Experiment 3: distorted visual space modifies object appearance

In the previous experiments, we found unique spatial distortions for individual observers (Experiment 1) and also discovered a potential source of the biases based on variations in visual acuity (Experiment 2). While acuity might be determined by early visual cortical processes [16], position

perception, arguably, emerges in extrastriate visual areas [18–20]. This hints at the possibility that idiosyncratic biases in perceived position (Experiment 1) might be inherited along the visual hierarchy to other later visual processing stages, such that they actually change the appearance of objects. We tested this question in Experiment 3 by measuring the perceived size of objects presented at different spatial positions.

## (a) Method
### (i) Participants
Three observers (1 female, 2 authors, age range: 23–33) who have participated in Experiment 1 and 2 participated in the current experiment. Experiment 1 and 2 have demonstrated that the association between acuity and spatial distortions is observer-specific, so a dense spatial sampling within a single subject will be necessary and sufficient to establish the relationship between variations in perceived size and spatial distortions. Therefore, in Experiment 3, we recruited fewer participants but with more locations tested for each participant. One observer was naïve to the purpose of the study. All subjects reported to have normal or corrected-to-normal vision. Procedures were approved by the Institutional Review Board at University of California, Berkeley.

### (ii) Stimuli
Stimuli were generated with the same software and displayed on the same monitor as Experiment 1 and 2. Observers were tested binocularly with a viewing distance of 40 cm fixed by a chin rest. Stimuli consisted of 'arcs' drawn from an invisible circle with a radius of 6 d.v.a. (figure 3a). The arc stimuli were shown at one of 20 angular positions equally distributed on the circle. Angular positions were separated by 18°, starting from 9°. There were six possible arc measures: 16.82°, 16°, 15.55°, 14.45°, 14°, 13.18°; the mean arc measure was 15°. Arc width was four arcmin of visual angle.

### (iii) Procedure
We used the method of single stimuli [21,22] to estimate the perceived size of objects across the visual field. Observers were instructed to maintain fixation throughout each trial on a black ($<0.001$ cd m$^{-2}$) 0.3 d.v.a. diameter fixation dot at the centre of a grey (48.3 cd m$^{-2}$) screen. After 1000 ms, a white (92.6 cd m$^{-2}$) arc (see descriptions above), with a pseudo-random length selected from the six possible lengths, was presented at one of the 20 predetermined positions. The arc was displayed for 500 ms and then it disappeared from the screen. Upon its offset, observers pressed either the left or right arrow key to indicate whether the presented arc was shorter or longer than the average of all seen arcs, regardless of stimulus location. Observers had unlimited time to make the key-press response and they were told to try to be as accurate as possible.

Observers were exposed to all possible arc lengths during a practice block, which was completed before the start of the experiment. The procedure in the practice was almost identical to the actual experiment, except that there were only 120 trials with each of the 6 arc lengths repeated 20 times. Also, observers received feedback in the practice block to speed their learning of the set mean; their accuracy was around

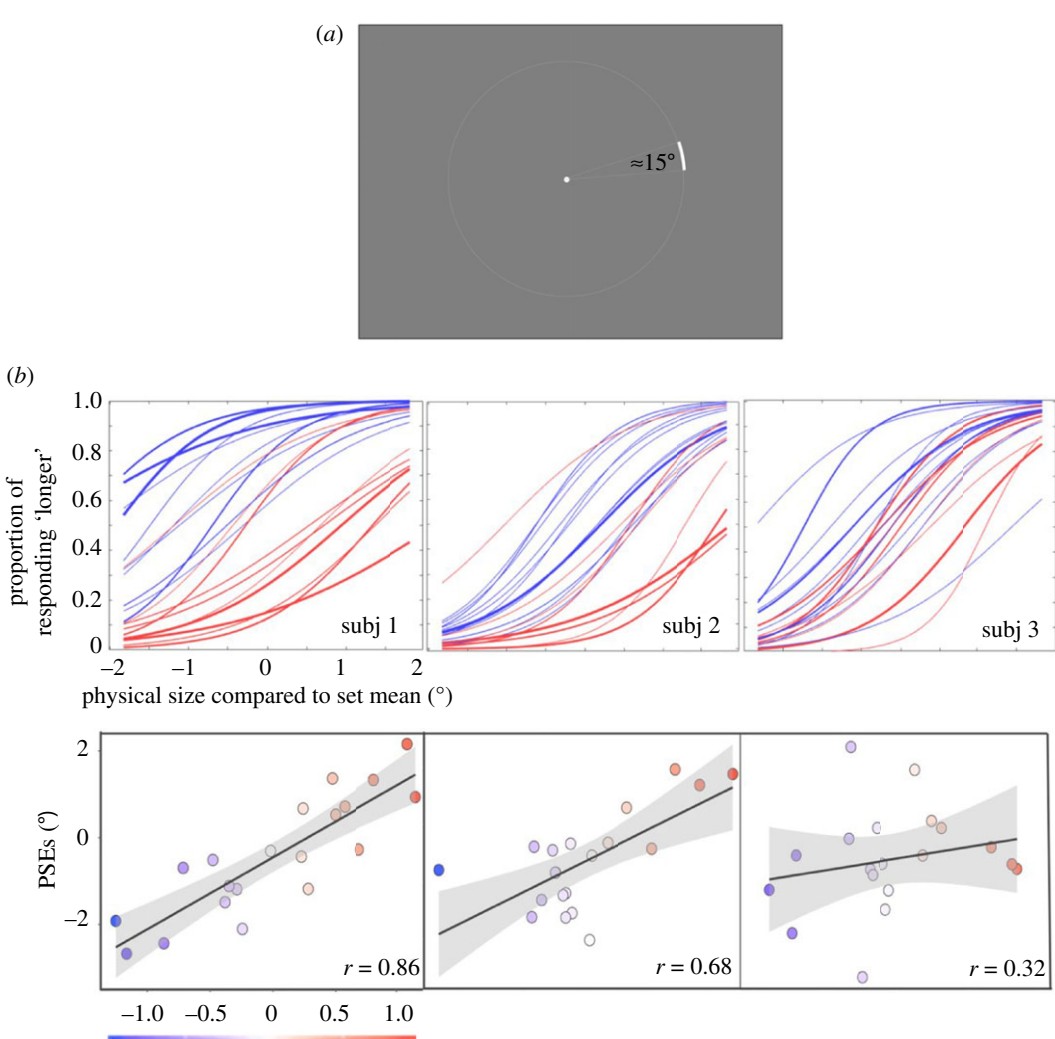

**Figure 3.** Experiment 3 paradigm and results. (*a*) Example arc stimulus used in the experiment. On each trial, an arc was presented at one of the 20 locations separated by 18° at an eccentricity of 6 d.v.a. Upon the offset of the arc, observers responded whether it was shorter or longer than the average. (*b*) Top: psychometric curves fitted for all three observers (Subjects 2 and 3 are authors). The abscissa represents the angular size difference between the presented and the mean arc. Colours of the lines correspond to different distortions obtained from individual subjects in Experiment 1 (blue for compression and red for expansion) and the weights of the curves correspond to the intensity of the distortion. Bottom: the association between the idiosyncratic visual space distortions (abscissa, from Experiment 1) and perceived size (ordinate) for each subject. The colour scale corresponds to the extent of compression or expansion quantified by the spatial distortion indices in Experiment 1. The positive correlations indicate that in regions of contracted (expanded) visual space, objects were perceived to be larger (smaller) than their actual size. The grey-shaded area around the regression line demonstrates the 95% CI of the linear regression fit. (Online version in colour.)

80% correct at the end of the practice session. In the actual experiment, no feedback was provided.

The whole experiment was divided into two sessions. In each session, observers finished a practice block and then an experiment block. In each experiment block, there were 1200 trials in total, with the condition that each possible length (6 in total) was presented 10 times at each possible location (20 in total). Observers were encouraged to take a rest after each set of 150 trials. The two sessions of 1200 trials each were collected on separate days, for a total of 2400 trials.

### (iv) Data analysis
To estimate the perceived size of the arc at each location, we calculated the proportion of trials in which the observer responded 'longer than average' for each length of the arc. Then these proportions were fitted to a logistic function using a least-squares procedure. The point of subjective

equality (PSE), which represents the perceived set mean, was defined as the arc length at which the proportion of 'longer' responses was 50% on the best-fit logistic function. A larger PSE represents a smaller perceived object size. We repeated this procedure for 20 tested locations; thus, we obtained 20 PSE values for each participant (figure 3). The change of PSE values as a function of different angular locations for each individual observer is shown in the electronic supplementary material, figure S4. To confirm the test–retest reliability, we estimated these 20 PSE values separately for each of the two sessions. The PSEs across sessions were averaged to test the relationship between the spatial distortions (Experiment 1) and the perceived object size (Experiment 3).

For each observer, each of the 20 locations in the current experiment was rounded to the two nearest corresponding locations tested in Experiment 1. The corresponding averaged distortion indices from Experiment 1 were correlated with the PSEs obtained in this experiment. To test whether the

correlation yielded for each observer was significantly higher than chance, we performed a permutation test. For each observer, we randomly shuffled the location labels of the PSEs and then correlated the shuffled PSEs with the distortion indices. This procedure was repeated 10 000 times to generate a null distribution and estimate the significance of the empirical correlation for each observer relative to this null distribution.

We analysed the Pearson's correlation between the PSEs from same locations but from different sessions to estimate the within-observer consistency of the spatial heterogeneity in perceived size for every participant. Between-observer consistency was also estimated by calculating pairwise correlations and averaging across all possible pairs.

## (b) Results

We found that across all three observers, spatial distortion indices were closely associated with perceived size (figure 3), with Pearson's correlations of 0.86, 0.68 and 0.32, respectively, for three observers ($p < 0.001$, $p < 0.001$ and $p = 0.13$ respectively, permutation test; Fisher's combined probability test: $p < 0.001$). This suggests that object appearance is altered depending on the local distortions in spatial coding. At perceptually compressed regions of the visual field (i.e. smaller distortion indices), objects are perceived to be larger, while the apparent size of objects is smaller at apparently expanded regions of the visual field. The within-observer consistency estimated by correlating PSEs across sessions, within each observer, showed that each observer had a stable (Pearson's correlations: 0.93, 0.94, 0.87, $ps < 0.001$, last two are the authors) and observer-specific (compared to a much lower between-observer, $r = 0.39$) spatial heterogeneity in the perceived sizes of objects throughout the visual field. Apparent object size is therefore affected by the unique spatial distortions across each observer's visual field, suggesting that idiosyncratic spatial biases are inherited along the visual hierarchy.

## 5. General discussion

In the present study, we found that observers are characterized by idiosyncratic spatial distortions, and we showed that these biases may arise from inhomogeneous spatial acuity across the visual field. These biases are then inherited along the visual hierarchy such that they change the appearance of object size. In Experiment 1, we found that different regions of the visual field in different observers are either effectively compressed or expanded. We demonstrated that these spatial distortions for each individual observer cannot be simply attributed to common biases among individuals. To trace the possible source of the distortions, Experiment 2 tested whether Vernier acuity varies at regions of the visual field that are perceptually compressed versus expanded. The results supported our hypothesis that spatial acuity is tightly linked to subject-specific spatial distortions. In particular, better acuity is associated with spatial compression and worse acuity is associated with a perceptual expansion of space. We also found that the association between acuity and the spatial distortion indices is observer-specific, which confirms the idiosyncratic nature of the spatial distortions. In Experiment 3, we asked whether the same idiosyncratic spatial biases can be inherited at higher levels of visual

processing, such that they change the appearance of visual objects. We found a stable and observer-specific spatial heterogeneity of perceived size of objects, which can be predicted by the idiosyncratic spatial distortions. Within regions of the visual field that were distinctly distorted (measured from biased position perception from Experiment 1), the apparent size of objects changed predictably: objects were perceived to be larger at locations with perceptually contracted spatial representation and smaller at expanded locations.

The individual differences in spatial distortions and apparent object size do not contradict known between-subject spatial biases or spatial anisotropies of vision such as the oblique effect [23]. In fact, we did find a significant between-observer correlation in Experiment 1. Also, when we averaged the idiosyncratic distortions across observers, we replicated other common spatial biases that have been reported by researchers [5–7]. For example, on average, observers tend to perceive visual space as expanded along the vertical meridian and compressed along the horizontal meridian (for average distortion map, see electronic supplementary material, figure S5). Since we found that better acuity underlies compressed visual areas and relatively worse acuity at expanded regions, this group-level result aligns with previous studies that showed a horizontal–vertical anisotropy [7]. Note that even along the vertical meridian, our results revealed an upper–lower visual field asymmetry, with the vertical axis in the upper visual field being more expanded. This also replicates the vertical meridian asymmetry in both human and non-human primates [5,24]. A more recent study carried out by Greenwood et al. [25] found that crowding zones and saccadic error zones are on average larger along the vertical than horizontal meridian and that there are variations between observers, which is consistent with what we showed in the current study. Our study goes beyond previous ones by showing that the individual differences in acuity, perceived position and perceived size are unique to the individual observer, are highly reliable and are not due to a group-level effect. Therefore, we believe that our current study reveals a fundamental idiosyncrasy underlying each observer's visual system and that our findings can potentially explain part of the common spatial biases reported in past research.

Despite the shared, between-subject, distortions and consistency (electronic supplementary material, figure S1), it is worth noting that the within-subject effects are sufficiently consistent and strong that they can swamp the between-subject effect. The within-subject correlation is not only higher (figure 1) but also contributed more unique variance than shared variance when explaining each subject's individual distortion indices (Experiment 1, Results). Experiment 2 showed that the association between distortions and Vernier acuity is significantly observer-specific (Experiment 2, Results). This suggests that individual differences in spatial resolution, as measured by Vernier acuity, are also substantial (electronic supplementary material, figure S3). Experiment 3 again revealed an observer-specific spatial heterogeneity of perceived size, which is reliably predicted by the idiosyncratic spatial distortions within each observer (Experiment 3, Results).

The within-subject effects also appear to be fairly stable over time, consistent with prior work [4]. The time span between Experiment 1 and 2 was 1–2 months, and the time span between Experiment 1 and 3 was around 11 months. Given the link we found between the biases estimated from

different experiments, this suggests some degree of stability in the within-subject idiosyncrasies, which are consistent with the temporally stable spatial distortions that were previously reported [4]. Overall, the rich variation between observers is in line with previous studies that showed substantial and stable idiosyncratic biases in the recognition of objects and patterns and motor decisions [26–29].

What causes the idiosyncratic acuity found in Experiment 2? One possibility is that variations in visual acuity arise from spatial inhomogeneities in early visual cortical processing. Previous studies have revealed that there are individual differences in the anatomical structure of human primary visual cortex [30], and that these are associated with differences in neural population tuning and perceptual performance in different visual tasks [31,32]. Therefore, it is plausible that even at the same eccentricity within individual observers, anatomical profiles corresponding to distinct regions of the visual field may be fundamentally heterogeneous and this creates an intrinsic spatial bias from early stages of the cortical visual hierarchy. When visual information is passed to later visual areas, local variations are inherited and neurons in those higher level visual regions can be impacted through neural undersampling processes [27]. Although speculative, our results suggest that this inheritance happens both in the ventral occipitotemporal visual pathway for object vision and the parietal pathway for spatial vision [33]; this includes higher level regions such as inferior temporal cortex, which has been linked to the perceived size of objects [34], and the posterior parietal cortex, which is responsible for spatial representation and head-centred localization [35].

How can the fundamental anatomical structure of our primary visual cortex be inhomogeneous? One possibility is random variation in receptive field number, size and/or density in any isoeccentric location of the visual field. Another possibility is that inhomogeneities are introduced through development. For example, heterogeneous spatial resolutions could be the aftermath of infant astigmatism [36]. Although significant astigmatism found in infancy typically disappears with age [37], such early astigmatism could introduce or shape inhomogeneities in visual cortical density and/or receptive field properties. It is also plausible that the idiosyncrasy we find may be determined naturally by genes and shaped by neuron migration, cortical connectivity and brain folding in early stages of life. For example, individual differences in the ability to recognize faces were found to be largely genetically driven through twin studies [38]. Future research on the visual development of individuals is needed to understand why humans develop such an idiosyncrasy in vision.

Although stable, consistent and accurate spatial localization is frequently assumed to be a simple product of retinotopic position, our results challenge this belief and demonstrate that every individual is characterized by idiosyncratically distorted visual space. These distortions may be induced by heterogeneous spatial acuity across the visual field and can influence visual appearance of object size, which suggests that these idiosyncratic fingerprints may be driven by variations in early visual cortex and are inherited along the visual hierarchy.

**Ethics.** All experimental procedures were approved by the Committee for the Protection of Human Subjects at the University of California, Berkeley.

**Data accessibility.** Supporting data can be found in the electronic supplementary material, and also by contacting the corresponding author.

**Authors' contributions.** All authors contributed to the study concept and the study design. Z.W. programmed the software and performed data collection and data analysis. The manuscript was drafted by Z.W. and was reviewed and edited by Y.M. and D.W. The figures were made by Z.W.

**Competing interests.** The authors declare no competing interests.

**Funding.** This work was supported in part by the National Institute of Health (grant no. 1R01CA236793-01) to D.W. and Grant-in-Aid for JSPS Fellows (grant no. 19J00039) to Y.M.

**Acknowledgment.** The authors are grateful for helpful discussions with Ken Nakayama, William Prinzmetal and Ervin Hafter. The authors also want to thank Connie Huang and Alexis Juarez for their assistance in data collection and data analysis.

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
