## [Reviewer comments · Proceedings of the Royal Society B: Biological Sciences]

Review History

RSPB-2020-0825.R0 (Original submission)

Review form: Reviewer 1

Recommendation

Accept with minor revision (please list in comments)

Scientific importance: Is the manuscript an original and important contribution to its field?

Excellent

General interest: Is the paper of sufficient general interest?

Good

Quality of the paper: Is the overall quality of the paper suitable?

Excellent

Is the length of the paper justified?

Yes

Should the paper be seen by a specialist statistical reviewer?

No

Do you have any concerns about statistical analyses in this paper? If so, please specify them explicitly in your report.

No

It is a condition of publication that authors make their supporting data, code and materials available - either as supplementary material or hosted in an external repository. Please rate, if applicable, the supporting data on the following criteria.

Is it accessible?

Yes

Is it clear?

No

Is it adequate?

No

Do you have any ethical concerns with this paper?

No

Comments to the Author

This study builds on this lab's previous work on idiosyncrasies in stimulus localisation biases across the visual field. They report that areas of the visual field (which broadly appear to be whole wedge-shaped regions) where position biases are pronounced (Exp 1) are associated with poorer Vernier acuity (Exp 2) and also with more reduction in perceived size (Exp 3). These findings are interesting in that they link previously reported idiosyncratic perceptual fingerprints and suggest a common neural substrate for them. This is a high quality study and I only have a few minor comments.

1. Page 15, Line 283-287: The behavioural responses in the Vernier task may have been confusing to participants. When the stimuli were near the upper-right quadrant the stimulus-response mapping is intuitive: A counter-clockwise displacement of the outer line is to the left of the inner line, and this requires subjects to respond with the Left key. However, when the stimuli are in the polar opposite location this mapping is reversed: Here a counter-clockwise replacement is to the right of the inner line, but the require response is still the Left key. In my experience, the intuitiveness of stimulus-response mappings can be problematic. Such an asymmetry would be reflected in differences in performance that in turn could skew the results. However, I don't think this is likely to have been a major problem here because the observers were all trained in psychophysics and these things can certainly be learned. It should be easy to rule this out by inspecting the lapse rates, that is, how much variability there was in performance at the easiest Vernier displacements. Moreover, this effect should really occur mostly in the lower-left quadrant.

2. Line 309: What exactly do you mean by collapsing data into a super-subject? Does this mean a long pooled vector where each entry is a position from one given observer? An alternative approach would be to establish the correlation between measures for each observer first, and then establish the ground average correlation is different from zero. There are advantages and disadvantages to either approach and I am not suggestion which one the authors should use as the results are typically fairly consistent. But please clarify.

3. Sample sizes: Why were there only 3 observers in Exp 3? I don't think this is a problem as the results are pretty clear and you can treat each observer as an independent replication. But it is at odds with the other experiments so could probably use some explanation.

4. Terminology: You refer to low spatial distortion as contraction and high distortion as expansion. This makes sense to me but I think that terminology may be confusing, especially later

when you discuss the relationship to perceived size where *expansion* results in *smaller* perceived size. To avoid confusion it would help to signpost this more and explicitly define this again at this point.

5. Figure 1: Panel label C is missing from the figure itself.

Sam Schwarzkopf
University of Auckland

Review form: Reviewer 2

Recommendation

Major revision is needed (please make suggestions in comments)

Scientific importance: Is the manuscript an original and important contribution to its field?

Excellent

General interest: Is the paper of sufficient general interest?

Good

Quality of the paper: Is the overall quality of the paper suitable?

Good

Is the length of the paper justified?

Yes

Should the paper be seen by a specialist statistical reviewer?

No

Do you have any concerns about statistical analyses in this paper? If so, please specify them explicitly in your report.

No

It is a condition of publication that authors make their supporting data, code and materials available - either as supplementary material or hosted in an external repository. Please rate, if applicable, the supporting data on the following criteria.

Is it accessible?

Yes

Is it clear?

Yes

Is it adequate?

Yes

Do you have any ethical concerns with this paper?

No

Comments to the Author

The authors investigated idiosyncratic distortions of spatial vision in three experiments by measuring perceived position, acuity and perceived size at multiple locations in the visual field. The results show idiosyncratic and correlated distortions in all three tasks, which points to a

common origin. The measurements are very extensive and carefully conducted. I have only a few comments for improving the presentation of the results in the manuscript.

Presentation of individual biases: For Vernier acuity and perceived size, only the correlation with the spatial distortions from Experiment 1 is shown in Figures 2 and 3. I think it would be useful to show Vernier acuity and perceived size as a function of location for all observers to get a full picture of the spatial variations in those tasks as well.

Stability of biases: The authors point out that they found stable biases at several places in the manuscript. While I tend to agree with that assessment, I think the time span over which stability was assessed should be stated more clearly.

Minor comments

Stimuli in Experiment 1: The luminance of the background is not stated in the manuscript.

Line 78: The authors state that the monitor frame was covered by black tape to minimize the influence of references. This would make sense if the background of the monitor was black, but Figure 1a suggests that it was actually gray.

Figure 1b: I suggest using a color map with continuously increasing luminance. Positive and negative values are indistinguishable in gray scale print with the current color map.

Line 253: The authors might want to mention that Vernier acuity unlike other acuity measurements exceeds the spatial resolution limits imposed by the maximal cone density on the retina (thus also called hyperacuity, Westheimer, 1975) and therefore tests acuity at a cortical rather than a retinal level.

Line 328: I recommend to provide degrees of freedom of the test statistics and to specify exact p-values whenever they are larger than 0.001. Was that correlation calculated for all locations and for all observers? This might lead to the problem of pseudoreplication, where multiple observations on the same observers are treated as independent samples (Lazic, 2010).

Lines 448 and 452: The caption of Figure 3 refers to left and right with respect to psychometric functions and correlation plots, but the arrangement in the figure is rather top and bottom.

Line 490: With respect to the asymmetry along the vertical meridian, the authors might want to relate to the seminal theory by Previc (1990).

Line 511: There are also idiosyncratic biases in motor decisions (e.g. Schütz, 2014).

References

Lazic, S. E. (2010). The problem of pseudoreplication in neuroscientific studies: is it affecting your analysis?. *BMC neuroscience*, 11(1), 5.

Previc, F. H. (1990). Functional specialization in the lower and upper visual fields in humans: Its ecological origins and neurophysiological implications. *Behavioral and Brain Sciences*, 13(3), 519-542.

Schütz, A. C. (2014). Inter-individual differences in preferred directions of perceptual and motor decisions. *Journal of Vision*, 14(12):16, 1-17.

Westheimer, G. (1975). Visual acuity and hyperacuity. *Investigative Ophthalmology & Visual Science*, 14(8), 570-572.

Decision letter (RSPB-2020-0825.R0)

12-May-2020

Dear Miss Wang:

Your manuscript has now been peer reviewed and the reviews have been assessed by an Associate Editor. The reviewers' comments (not including confidential comments to the Editor) and the comments from the Associate Editor are included at the end of this email for your reference. As you will see, the reviewers and the Editors have raised some concerns with your manuscript and we would like to invite you to revise your manuscript to address them.

Research ethics:

Use of animals and field studies:

Please submit a copy of your revised paper within three weeks. If we do not hear from you within this time your manuscript will be rejected. If you are unable to meet this deadline please let us know as soon as possible, as we may be able to grant a short extension.

Best wishes,
Dr Robert Barton
mailto: proceedingsb@royalsociety.org

Associate Editor
Comments to Author:

We have now heard from two experts. I am pleased to say that both are enthusiastic about your manuscript. Nevertheless, they have raised some issues that you will have to deal with before we can move forward. I invite you to submit a revised manuscript.

Reviewer(s)' Comments to Author:

Referee: 1

Comments to the Author(s)

This study builds on this lab's previous work on idiosyncrasies in stimulus localisation biases across the visual field. They report that areas of the visual field (which broadly appear to be whole wedge-shaped regions) where position biases are pronounced (Exp 1) are associated with poorer Vernier acuity (Exp 2) and also with more reduction in perceived size (Exp 3). These findings are interesting in that they link previously reported idiosyncratic perceptual fingerprints

and suggest a common neural substrate for them. This is a high quality study and I only have a few minor comments.

1. Page 15, Line 283-287: The behavioural responses in the Vernier task may have been confusing to participants. When the stimuli were near the upper-right quadrant the stimulus-response mapping is intuitive: A counter-clockwise displacement of the outer line is to the left of the inner line, and this requires subjects to respond with the Left key. However, when the stimuli are in the polar opposite location this mapping is reversed: Here a counter-clockwise replacement is to the right of the inner line, but the require response is still the Left key. In my experience, the intuitiveness of stimulus-response mappings can be problematic. Such an asymmetry would be reflected in differences in performance that in turn could skew the results. However, I don't think this is likely to have been a major problem here because the observers were all trained in psychophysics and these things can certainly be learned. It should be easy to rule this out by inspecting the lapse rates, that is, how much variability there was in performance at the easiest Vernier displacements. Moreover, this effect should really occur mostly in the lower-left quadrant.

2. Line 309: What exactly do you mean by collapsing data into a super-subject? Does this mean a long pooled vector where each entry is a position from one given observer? An alternative approach would be to establish the correlation between measures for each observer first, and then establish the ground average correlation is different from zero. There are advantages and disadvantages to either approach and I am not suggestion which one the authors should use as the results are typically fairly consistent. But please clarify.

3. Sample sizes: Why were there only 3 observers in Exp 3? I don't think this is a problem as the results are pretty clear and you can treat each observer as an independent replication. But it is at odds with the other experiments so could probably use some explanation.

4. Terminology: You refer to low spatial distortion as contraction and high distortion as expansion. This makes sense to me but I think that terminology may be confusing, especially later when you discuss the relationship to perceived size where *expansion* results in *smaller* perceived size. To avoid confusion it would help to signpost this more and explicitly define this again at this point.

5. Figure 1: Panel label C is missing from the figure itself.

Sam Schwarzkopf
University of Auckland

Referee: 2

Comments to the Author(s)

The authors investigated idiosyncratic distortions of spatial vision in three experiments by measuring perceived position, acuity and perceived size at multiple locations in the visual field. The results show idiosyncratic and correlated distortions in all three tasks, which points to a common origin. The measurements are very extensive and carefully conducted. I have only a few comments for improving the presentation of the results in the manuscript.

Presentation of individual biases: For Vernier acuity and perceived size, only the correlation with the spatial distortions from Experiment 1 is shown in Figures 2 and 3. I think it would be useful to show Vernier acuity and perceived size as a function of location for all observers to get a full picture of the spatial variations in those tasks as well.

Stability of biases: The authors point out that they found stable biases at several places in the manuscript. While I tend to agree with that assessment, I think the time span over which stability was assessed should be stated more clearly.

Minor comments

Stimuli in Experiment 1: The luminance of the background is not stated in the manuscript.

Line 78: The authors state that the monitor frame was covered by black tape to minimize the influence of references. This would make sense if the background of the monitor was black, but Figure 1a suggests that it was actually gray.

Figure 1b: I suggest using a color map with continuously increasing luminance. Positive and negative values are indistinguishable in gray scale print with the current color map.

Line 253: The authors might want to mention that Vernier acuity unlike other acuity measurements exceeds the spatial resolution limits imposed by the maximal cone density on the retina (thus also called hyperacuity, Westheimer, 1975) and therefore tests acuity at a cortical rather than a retinal level.

Line 328: I recommend to provide degrees of freedom of the test statistics and to specify exact p-values whenever they are larger than 0.001. Was that correlation calculated for all locations and for all observers? This might lead to the problem of pseudoreplication, where multiple observations on the same observers are treated as independent samples (Lazic, 2010).

Lines 448 and 452: The caption of Figure 3 refers to left and right with respect to psychometric functions and correlation plots, but the arrangement in the figure is rather top and bottom.

Line 490: With respect to the asymmetry along the vertical meridian, the authors might want to relate to the seminal theory by Previc (1990).

Line 511: There are also idiosyncratic biases in motor decisions (e.g. Schütz, 2014).

References

Lazic, S. E. (2010). The problem of pseudoreplication in neuroscientific studies: is it affecting your analysis?. *BMC neuroscience*, 11(1), 5.

Previc, F. H. (1990). Functional specialization in the lower and upper visual fields in humans: Its ecological origins and neurophysiological implications. *Behavioral and Brain Sciences*, 13(3), 519-542.

Schütz, A. C. (2014). Inter-individual differences in preferred directions of perceptual and motor decisions. *Journal of Vision*, 14(12):16, 1-17.

Westheimer, G. (1975). Visual acuity and hyperacuity. *Investigative Ophthalmology & Visual Science*, 14(8), 570-572.

Author's Response to Decision Letter for (RSPB-2020-0825.R0)

See Appendix A.

Decision letter (RSPB-2020-0825.R1)

15-Jun-2020

Dear Miss Wang

I am pleased to inform you that your manuscript entitled "Idiosyncratic Perception: A Link Between Acuity, Perceived Position and Apparent Size" has been accepted for publication in Proceedings B.

Open Access

Paper charges

Sincerely,

Dr Robert Barton

Associate Editor:

Board Member

Comments to Author:

I have good news. I am recommending that your manuscript be accepted for publication in Proceedings. It is an excellent set of studies. Congratulations.

Appendix A

Responses to Referees

Referee: 1

Comments to the Author(s)

This study builds on this lab's previous work on idiosyncrasies in stimulus localisation biases across the visual field. They report that areas of the visual field (which broadly appear to be whole wedge-shaped regions) where position biases are pronounced (Exp 1) are associated with poorer Vernier acuity (Exp 2) and also with more reduction in perceived size (Exp 3). These findings are interesting in that they link previously reported idiosyncratic perceptual fingerprints and suggest a common neural substrate for them. This is a high quality study and I only have a few minor comments.

Response: Thank you very much for your positive review, careful reading, and constructive feedback. We have made each recommended change in the manuscript.

1. Page 15, Line 283-287: The behavioural responses in the Vernier task may have been confusing to participants. When the stimuli were near the upper-right quadrant the stimulus-response mapping is intuitive: A counter-clockwise displacement of the outer line is to the left of the inner line, and this requires subjects to respond with the Left key. However, when the stimuli are in the polar opposite location this mapping is reversed: Here a counter-clockwise replacement is to the right of the inner line, but the require response is still the Left key. In my experience, the intuitiveness of stimulus-response mappings can be problematic. Such an asymmetry would be reflected in differences in performance that in turn could skew the results. However, I don't think this is likely to have been a major problem here because the observers were all trained in psychophysics and these things can certainly be learned. It should be easy to rule this out by inspecting the lapse rates, that is, how much variability

there was in performance at the easiest Vernier displacements. Moreover, this effect should
really occur mostly in the lower-left quadrant.

Responses: Good point, and good idea. Following the advice above, we have confirmed that
there was no difference in lapse rates at different locations [No significant performance
difference between upper ($M = 96.43\%$, $SD = 4\%$) and lower ($M = 96.87\%$, $SD = 3\%$) visual
field ($F(1,52) < 0.2$, $p > .5$). There is no significant difference between left ($M = 97.32\%$, SD
$= 4\%$) and right ($M = 95.98\%$, $SD = 4\%$) visual field ($F(1,52) < 1.6$, $p > .2$). The interaction
between upper/lower and left/right visual field is not significant either ($F(1,52) < .6$, $p > .4$).
All participants were well-trained and experienced in psychophysics, as well.

2. Line 309: What exactly do you mean by collapsing data into a super-subject? Does this
mean a long pooled vector where each entry is a position from one given observer? An
alternative approach would be to establish the correlation between measures for each
observer first, and then establish the ground average correlation is different from zero. There
are advantages and disadvantages to either approach and I am not suggestion which one the
authors should use as the results are typically fairly consistent. But please clarify.

Responses: Thank you for asking about it. Our analysis of data from Experiment 2 originally
followed the first analysis that Referee 1 mentioned here. However, as Referee 2 suggested,
this analysis might be subject to the problem of pseudoreplication (Lazic, 2010). Therefore,
instead of analyzing the super-subject data, to separate multiple dependent observations from
each observer and independent observers, we performed the second analysis Referee 1
suggested here with additional analyses listed below.

- 1. We calculated the Pearson's correlation between spatial distortions and Vernier acuity
on every observer separately. This yielded 7 Pearson's r values, which were
transformed to Fisher z values, averaged together and then transformed back to
Pearson's r . This resulted in an average correlation of 0.34. We also performed a
bootstrap procedure on these correlation values. On each iteration, we randomly
sampled 7 correlation values with replacement from the 7 empirical correlation values
and applied a Fisher transformation on each sample. Then the 7 Fisher z values were
averaged together and transformed back to Pearson's r to estimate a mean
bootstrapped correlation among observers. We repeated this procedure for 1,000
59 times and estimated the 95% bootstrapped confidence interval of mean correlation
among observers. This additional analysis yielded a 95% bootstrapped confidence
interval of [0.06, 0.56], which suggested that the average correlation from different
observers is significantly different from 0. We have now included this additional
analysis in the main manuscript.
- 2. We also fitted a linear mixed-effect model (which specifies the association between
spatial distortions and Vernier acuity as fixed effect and the inter-individual
difference as a random effect) to examine whether individual differences play a role
in the association between Vernier acuity and spatial distortions. The model results
suggested a significant and positive association between spatial distortions and
Vernier acuity, with a fixed effect coefficient of 0.63 (standard error: 0.23, $F(1, 48) =$
$7.59, p < .01$). We have also included the linear mixed-effect model results in the
main manuscript.
- 3. We believed that showing individual observer correlations will also be helpful, so we
now included individual observer correlations in the supplementary materials (Fig.
S3).

Figure S3. Correlation between spatial distortion indices and Vernier acuity JNDs for each observer. Each observer had 8 pairs of data, corresponding to 8 angular locations tested in Experiment 2. Different symbols represent different observers. Lines are regression lines fitted based on each observer's data. The Pearson's correlations for individual subjects were 0.31, 0.73, 0.34, 0.55, -0.37, 0.26, 0.38 (listed in the same order as the figure legend) and the mean correlation calculated from Fisher transformation was 0.34. Note that the only observer who did not show the same trend (displayed as gray diamond) had the smallest JNDs (i.e., best acuity), so we speculated that it might be subject to a ceiling effect. This could affect the measured variability of Vernier acuity across different locations and thus influence the correlation calculated based on it.

3. Sample sizes: Why were there only 3 observers in Exp 3? I don't think this is a problem as the results are pretty clear and you can treat each observer as an independent replication. But

it is at odds with the other experiments so could probably use some explanation.

Responses: Our reasoning was that since Experiment 1 and 2 have made it clear that this
idiosyncratic association is observer-specific, we believed that a dense spatial sampling
within a single subject would be more helpful to establish the relationship between variations
in perceived size and heterogeneous spatial distortions. Therefore, in Experiment 3, we
recruited fewer participants but with more locations tested for each participant. We now
added our reasons in the Method section of Experiment 3 of the revised manuscript.

4. Terminology: You refer to low spatial distortion as contraction and high distortion as
expansion. This makes sense to me but I think that terminology may be confusing, especially
later when you discuss the relationship to perceived size where *expansion* results in
*smaller* perceived size. To avoid confusion it would help to signpost this more and
explicitly define this again at this point.

Responses: The reason why we used expansion and contraction to describe the spatial
distortion is based on our operational definition of them. Since a positive distortion index
indicates that two adjacent objects were localized to be further away from each other
compared to their actual physical distance, we believe that it indicated that the visual space
between these two locations were effectively expanded, and the opposite was true for a
negative distortion index. Thus, “expansion” and “contraction” were defined based on biased
perceived position rather than size perception. We revisited the definition of the terminology
in the discussion section to avoid confusion between perceived position and perceived size.

5. Figure 1: Panel label C is missing from the figure itself.

Responses: Thank you so much for catching this mistake and we have updated the figure to include the label “C”.

Referee: 2

Comments to the Author(s)

The authors investigated idiosyncratic distortions of spatial vision in three experiments by measuring perceived position, acuity and perceived size at multiple locations in the visual field. The results show idiosyncratic and correlated distortions in all three tasks, which points to a common origin. The measurements are very extensive and carefully conducted. I have only a few comments for improving the presentation of the results in the manuscript.

Response: Thanks for your thoughtful review and constructive comments. We have made all of the recommended changes to the manuscript.

Presentation of individual biases: For Vernier acuity and perceived size, only the correlation with the spatial distortions from Experiment 1 is shown in Figures 2 and 3. I think it would be useful to show Vernier acuity and perceived size as a function of location for all observers to get a full picture of the spatial variations in those tasks as well.

Responses: This is a great suggestion. We have included the change of Vernier acuity and
perceived size as a function of locations for each individual observer in the supplemental
figures (see supplemental figure S2 and S4).

Figure S2. Change of Vernier acuity as a function of the angular locations tested for every observer.
Subject 1 and Subject 4 are authors. The layout of the angular locations is shown on the bottom right
corner.

Figure S4. The change of perceived size of the arc stimuli as a function of the angular locations tested
for every observer. Subject 1 and Subject 2 are authors. The layout of the angular locations is shown
on the bottom right corner.

Stability of biases: The authors point out that they found stable biases at several places in the
manuscript. While I tend to agree with that assessment, I think the time span over which
stability was assessed should be stated more clearly.

Responses: Thank you very much for asking about it. Firstly, Kosovicheva & Whitney (2017)
tested the stability of the localization biases across time so we cited their paper at the
beginning of Experiment 1. To make it clearer, we mentioned this now in the revised
introduction part of the manuscript. Although the time span within each of our experiments is
limited (within a week), the time span between Experiment 1 and 2 was 1~2 months, and the
time span between Experiment 1 and 3 was ~11 months. Since we still found a stable
association between the biases estimated from different experiments, this indicates a kind of
stability and is consistent with the temporally stable spatial distortions that Kosovicheva &
Whitney (2017) reported. We have clarified this in the updated discussion section.

Minor comments

Stimuli in Experiment 1: The luminance of the background is not stated in the manuscript.

Responses: We reported the luminance of the gray background in the Procedure section in
Experiment 1. It was 48.3 cd/m^2 .

Line 78: The authors state that the monitor frame was covered by black tape to minimize the

influence of references. This would make sense if the background of the monitor was black,
but Figure 1a suggests that it was actually gray.

Responses: We have corrected the language in the methods to specify what we meant. The
black tape helped to minimize off-screen references: any visible references outside of the
computer monitor including the difference between the monitor frame and the experiment
room.

Figure 1b: I suggest using a color map with continuously increasing luminance. Positive and
negative values are indistinguishable in gray scale print with the current color map.

Responses: We have created a version of this of figure using added luminance gradients and
put in the supplemental material for reference (Fig. S1).

**Figure S1.** The gray-scale version of the spatial distortion maps reported in Experiment 1. Brighter
color (negative spatial distortion indices) indicates contraction of visual space and darker color
(positive spatial distortion indices) represents expanded visual space.

Line 253: The authors might want to mention that Vernier acuity unlike other acuity

measurements exceeds the spatial resolution limits imposed by the maximal cone density on
the retina (thus also called hyperacuity, Westheimer, 1975) and therefore tests acuity at a
cortical rather than a retinal level.

**Responses: Thank you for the suggestion and we added a reference to it at the beginning of**
**Experiment 2.**

Line 328: I recommend to provide degrees of freedom of the test statistics and to specify
exact p-values whenever they are larger than 0.001. Was that correlation calculated for all
locations and for all observers? This might lead to the problem of pseudoreplication, where
multiple observations on the same observers are treated as independent samples (Lazic,
2010).

**Responses: This is a really good concern. The correlation was originally calculated for all**
**locations across all observers (i.e., each entry is a position from one given observer) and this**
**analysis is indeed subject to the problem of pseudoreplication. Therefore, instead of**
**analyzing the super-subject data, to separate multiple dependent observations from each**
**observer and independent observers, we performed the following analyses instead.**

- 1. We calculated the Pearson's correlation between spatial distortions and Vernier
acuity on every observer separately. This yielded 7 Pearson's r values, which
were transformed to Fisher z values, averaged together and then transformed back
to Pearson's r . This resulted in an average correlation of 0.34. We also performed
a bootstrap procedure on these correlation values. On each iteration, we randomly
sampled 7 correlation values with replacement from the 7 empirical correlation
values and applied a Fisher transformation on each sample. Then the 7 Fisher z

values were averaged together and transformed back to Pearson's r to estimate a
mean bootstrapped correlation among observers. We repeated this procedure for
1,000 times and estimated the 95% bootstrapped confidence interval of mean
correlation among observers. This additional analysis yielded a 95% bootstrapped
confidence interval of [0.06, 0.56], which suggested that the average correlation
from different observers is significantly different from 0. We have now included
this additional analysis in the main manuscript.

- 2. We also fitted a linear mixed-effect model (which specifies the association
between spatial distortions and Vernier acuity as fixed effect and the inter-
individual difference as a random effect) to examine whether individual
differences play a role in the association between Vernier acuity and spatial
distortions. The model results suggested a significant and positive association
between spatial distortions and Vernier acuity, with a fixed effect coefficient of
0.63 (standard error: 0.23, $F(1, 48) = 7.59, p < .01$). We have also included the
linear mixed-effect model results in the main manuscript.
- 3. We believed that showing individual observer correlations will also be helpful, so
we now included individual observer correlations in the supplementary materials
(Fig. S3).

**Figure S3.** Correlation between spatial distortion indices and Vernier acuity JNDs for each observer.
 Each observer had 8 pairs of data, corresponding to 8 angular locations tested in Experiment 2.
 Different symbols represent different observers. Lines are regression lines fitted based on each
 observer's data. The Pearson's correlations for individual subjects were 0.31, 0.73, 0.34, 0.55, -0.37,
 0.26, 0.38 (listed in the same order as the figure legend) and the mean correlation calculated from
 Fisher transformation was 0.34. Note that the only observer who did not show the same trend
 (displayed as gray diamond) had the smallest JNDs (i.e., best acuity), so we speculated that it might
 be subject to a ceiling effect. This could affect the measured variability of Vernier acuity across
 different locations and thus influence the correlation calculated based on it.

Lines 448 and 452: The caption of Figure 3 refers to left and right with respect to

psychometric functions and correlation plots, but the arrangement in the figure is rather top

and bottom.

**Responses: Thank you for catching this error and we have corrected the caption now.**

Line 490: With respect to the asymmetry along the vertical meridian, the authors might want
to relate to the seminal theory by Previc (1990).

**Responses: Thank you for the suggestion and we also referred to it in our updated discussion**
**session now.**

Line 511: There are also idiosyncratic biases in motor decisions (e.g. Schütz, 2014).

**Responses: Thank you very much and we included this in our discussion section.**